# A Light/Pressure Bifunctional Electronic Skin Based on a Bilayer Structure of PEDOT:PSS-Coated Cellulose Paper/CsPbBr_3_ QDs Film

**DOI:** 10.3390/polym15092136

**Published:** 2023-04-29

**Authors:** Wenhao Li, Jingyu Jia, Xiaochen Sun, Sue Hao, Tengling Ye

**Affiliations:** 1CAS Key Laboratory of Renewable Energy, Guangzhou Institute of Energy Conversion, Guangzhou 510640, China; 2Department of Applied Chemistry, School of Chemistry and Chemical Engineering, Harbin Institute of Technology, Harbin 150001, China; liwenhao0824@163.com (W.L.); jjy271828@163.com (J.J.); 15684172397@163.com (X.S.)

**Keywords:** e-skin photodetector, pressure sensor, CsPbBr_3_, PEDOT: PSS, e-skin

## Abstract

With the continuous development of electronic skin (e-skin), multifunctional e-skin is approaching, and in some cases even surpassing, the capabilities of real human skin, which has garnered increasing attention. Especially, if e-skin processes eye’s function, it will endow e-skins more powerful advantages, such as the vision reparation, enhanced security, improved adaptability and enhanced interactivity. Here, we first study the photodetector based on CsPbBr_3_ quantum dots film and the pressure sensor based on PEDOT: PSS-coated cellulose paper, respectively. On the base of these two kinds of sensors, a light/pressure bifunctional sensor was successfully fabricated. Finally, flexible bifunctional sensors were obtained by using a flexible interdigital electrode. They can simultaneously detect light and pressure stimulation. As e-skin, a high photosensitivity with a switching ratio of 168 under 405 nm light at a power of 40 mW/cm^2^ was obtained and they can also monitor human motions in the meantime. Our work showed that the strategy to introduce perovskite photodetectors into e-skins is feasible and may open a new way for the development of flexible multi-functional e-skin.

## 1. Introduction

Electronic skin (e-skin) is an electronic system to simulate the function of human skin. It is a kind of important flexible wearable electronic device that has a promising prospect in the field of health monitoring, human-machine interaction, software robots, and the Internet of Things [1,2,3,4]. Recently, multifunctional e-skins have become a hot direction to fulfill the requirements of specific wearable and portable applications. In the many exploration processes to realize multifunctional e-skins, it is the main work to simulate the function of human skin, such as the sensory abilities to stress, strain, temperature, humidity and chemicals, etc. [5,6,7,8]. However, in addition to the function of human skin, there are many ways for human beings to obtain external information. The eyes, ears, nose, and mouth are indispensable channels for obtaining external information. Especially, for realizing the eye’s function, the retina is a key film-shaped tissue, which directly detects light and outputting/transmitting a bioelectricity signal to the nervous system [9]. If e-skins have the function of the retina, they may be endowed with some very interesting advantages [10]. For example, e-skins with visual perception can perceive light and images such as eyes, thus the user’s safety in dangerous environments can be improved, avoiding potential dangers, such as high-energy radiation damage, and improving environmental adaptability [11]. Through the visual perception ability of e-skins, the robot with these e-skins can better interact with humans and enhance the ability of human–machine interactions. In addition, it may be more powerful in future medical care and may play an important role in vision reparation for people who are blind [10]. Anyway, it will bring broader application prospects and more functional advantages.

Perovskite photodetectors have recently become popular due to their exceptional photoelectric properties and easy fabrication process. [12]. Inorganic halide CsPbBr_3_ perovskite quantum dots (PQDs) are considered as a new generation of photoelectric material for the photodetector due to their merits including a high light absorption coefficient, a long charge carrier life, a tunable bandgap, relatively high stability, inexpensiveness, and flexible compositional control [13,14]. With the deepening of research, more and more multifunctional photodetectors have been developed [15,16]. What is more, the absorption of CsPbBr_3_ QDs is in the high energy region from UV to blue-green, and it is suitable for the high-energy radiation detection. On the other hand, piezoresistive flexible pressure sensors (PFPSs) can transduce pressure into corresponding resistance signals and have been extensively investigated as e-skin owing to their low cost, simple assembly process and excellent sensing performances [17,18]. Generally, PFPSs usually consist of an elastic, porous, conductive interlayer sandwiched between two flexible electrodes. Porous cellulose paper is considered as a good candidate for the interlayer framework materials owing to its low cost, light weight, high surface area, outstanding deformability, excellent flexibility, and air permeability. When combining the porous cellulose paper with common conductive materials, such as carbon nanomaterials, metal nanomaterials or conductive polymers, highly sensitive FPSs with a low cost can be easily obtained. However, the conductive materials in these FPSs are usually opaque, which cannot meet the requirement of light sensing e-skins [19,20].

In this paper, we propose to combine a perovskite photodetector with a piezoresistive pressure sensor to design a bifunctional sensor with both light and pressure response. The light/pressure bifunctional e-skins were prepared based on PEDOT: PSS-coated cellulose paper/CsPbBr_3_ QDs film. Firstly, we studied the photodetectors based on CsPbBr_3_ PQDs and optimized the fabrication process on the rigid conductive glass. Then, the transparent conductive polymer PEDOT: PSS and cellulose papers were selected to fabricate the piezoresistive pressure sensor. After that the two sensors are integrated together to prepare the light/pressure bifunctional sensors. Finally, the flexible bifunctional sensors are successfully obtained by replacing the rigid conductive glass with a flexible interdigital electrode. In addition, flexible bifunctional sensors are successfully applied to human physiological signal monitoring.

## 2. Materials, Methods, and Characterization

### 2.1. Materials

PbBr_2_ (99%) were supplied by Shanghai Energy Chemical Co., Ltd., Shanghai, China. and Cs_2_CO_3_ (99.9%) were purchased from ZhongNuo Advanced Material (Beijing) Technology Co., Ltd., Beijing, China; Oleamine (OLA with 90%) and Octadecylene (90%) were purchased from Shanghai Macklin Biochemical Technology Co., Ltd., Shanghai, China; Oleic acid (OA, analytical reagent) was supplied by DaMao Chemical Reagent Factory, Tianjin, China; n-Hexane, Isopropyl alcohol and acetone (analytical pure) were purchased from Tianjin Fuyu Chemical Reagent Co., Ltd., Tianjin, China; PEDOT:PSS, 4083, was purchased from Xi’an Bao Laite Optoelectronics Technology Co., Ltd., Xi’an, China; Cellulose paper (plain weave, 60 g, 27.5 cm × 57.5 cm) was supplied from Sichuan Sanheng Yishu Technology Co., Ltd., Chengdu, China.

### 2.2. Methods

#### 2.2.1. Synthesis of CsPbBr_3_ PQDs

The synthesis follows the modified method reported by Wang et al. [21]. 0.65 g Cs_2_CO_3_, 2.5 mL of OA, and 18 mL of ODE were mixed and kept in an N_2_ atmosphere. The mixture was heated up to 130 °C for 1 h. After that, the mixture was further heated up to 150 °C for reaction and lasted for 0.5 h. Then, the solution was cooled down to room temperature. On the other side, 2 mL ODE, 0.3 mL OA, 0.3 mL OLA and 0.2 mmol of PbBr_2_, were put together and kept in an N_2_ atmosphere for 1 h at temperature of 120 °C. Then, the solution was further heated up to 160 °C for reaction for 10 min. Finally, the Cs precursor (0.2 mL) was quickly added into the solution and cooled down with an ice bath. After centrifugation, the resulting pellet was dispersed with n-hexane.

#### 2.2.2. Preparation of Photodetectors Based on CsPbBr_3_ QDs Film

ITO glass with a slit was firstly washed with alkaline dish soap to remove dust from the surface. After that, three alternating ultrasounds were used with acetone and isopropanol for 30 min each time. Next, a certain amount of washing solvent (isopropanol or acetone) is added into the obtained QDs solution in n-hexane according to the n-hexane/solvent volume ratio of 1:3. Then, we repeat the process of centrifugation and dispersion, and the washed CsPbBr_3_ was redispersed in n-hexane. Finally, the indium tin oxide glass (ITO) is soaked in the washed CsPbBr_3_ solution, and a CsPbBr_3_ film was obtained by the centrifugation method (5000 r/min, 20 mg/mL). The CsPbBr_3_ film on ITO was placed on a hot plate at 120 °C for 10 min to evaporate the excess solvent. ITO is 1.5 cm × 1.0 cm in size, and there is an etched area with a width of 105 μm.

#### 2.2.3. Preparation of Pressure Sensors Based on PEDOT: PSS-Coated Cellulose Paper

The spunlace and needled non-woven paper are made of polyethylene terephthalate (PET) and cellulose paper was made of cellulose. All of them were cut into the size of 0.6 cm × 1 cm. Next, different volumes of PEDOT: PSS solution were dropwise added to the paper, and then they were dried in an oven at 80 °C for 20 min. The PEDOT: PSS-coated cellulose paper was placed on top of the ITO glass. Finally, a polydimethylsiloxane (PDMS) film is used to package and fix the pressure sensor with the structure of PEDOT: PSS-coated cellulose paper/ITO.

#### 2.2.4. Preparation of Light/Pressure Bifunctional Sensors

Firstly, the CsPbBr_3_ QDs film on ITO was fabricated using the centrifugation method as described in 2.2.2. After that, 20 μL of PEDOT: PSS aqueous solution was added dropwise to the cellulose paper (0.6 cm × 1 cm) which was dried in an oven at 80 °C for 20 min. Next, the conductive paper was placed on top of the ITO glass. Finally, the PDMS film was used to package and fix the bifunctional sensor. For the flexible bifunctional sensors, we just simply replaced the ITO glass with a flexible interdigital electrode (Au/polyimide, 10 mm × 20 mm × 13 μm, Au width: 100 μm, Au distance: 100 μm, 20 pairs of electrodes and the electrode length is 6.3 mm).

### 2.3. Characterization

The morphology of CsPbBr_3_ QDs were performed on a transmission electron microscope (TEM, JEM 1400 plus, 100 kV). The Fourier transform infrared (FTIR) spectrum of CsPbBr_3_ QDs was recorded by a Fourier infrared spectrometer (FTIR-7600, Lambda). X-ray diffraction (XRD) patterns in the range of 5–85° were recorded on an PANalytical X’Pert Powder diffractometer using Cu-Kα radiation (λ = 1.5406 Å) (PANalytical, Almelo, The Netherlands). Optical microscope photographs were obtained by using a TXS06-02H Fluorescence Microscope (Srate Optical Instrument Manufactory, Nanyang, China). The absorption spectrum of CsPbBr_3_ QDs were measured by a Ultraviolet-visible spectrophotometer (Cary 100, Agilent Technologies Inc., Santa Clara, CA, USA) and the fluorescence spectrums were measured using an optic fiber associated with a spectrometer (Ocean Optics USB 4000). The samples were dissolved in n-hexane with a concentration of 1 × 10^−4^ mol/L. Furthermore, the pressure sensing performances of the pressure sensor were implemented on a homemade test system. A 50 g weight was connected to a dip coater to apply reciprocating pressure on pressure sensors and a KEITHLEY 2400 source meter connected to a computer was used for the in-situ electrical signal collection during the expansive and compressive process. The current–voltage curves of the photodetector and the pressure sensors were also recorded using the KEITHLEY 2400 source meter; the applied bias is 1 V or 10 V. For photoelectric measurement, the excitation light source was an ultra-violet source with a 365 filter (EXECURE 4000, HOYA, Tokyo, Japan) and a LED light source (405 nm, 532 nm, 660 nm), Guangzhou Jingyi Optoelectronic Technology Co., Ltd.). When the LED light source of JY18102301 is used, the distance between the source and photodetector was determined by the power densities which were estimated using a power meter (Newport, Model 2936-C). The switching ratio of the photo detector was calculated using the switching ratio = (I_p_/I_d_) = ((I_on_ − I_d_)/(I_d_)).

## 3. Results and Discussion

### 3.1. Preparation and Properties of PQDs Photodetector

CsPbBr_3_ QDs were successfully synthesized by the typical hot injection method [22,23] and the morphology and structure can be confirmed by the TEM and XRD. The TEM image of CsPbBr_3_ QDs is shown in Figure 1a and the average size of CsPbBr_3_ QDs is 10~15 nm. The XRD pattern of CsPbBr_3_ QDs in Figure 1b shows that its structure belongs to cubic phase (JC//PDS No. 54-0752). As described in the introduction, the CsPbBr_3_ QDs may be well applicable for high energy photodetectors because of their range of light absorptivity and high stability. As shown in Figure 1c and Appendix A, The UV–vis absorption spectrum indicates that the CsPbBr_3_ QDs have a direct bandgap of about 2.49 eV and the fluorescence spectrum in Figure 1d shows an emission peak at 521 nm. Compared to the bulk crystal, the decrease in size of the CsPbBr_3_ QDs results in a blue-shift of the absorption onset due to quantum size effects [24,25]. In addition, the absorbance range indicates that the CsPbBr_3_ QDs can be used as the active layer for UV-blue-green photodetectors.

The planar photodetector device based on CsPbBr_3_ QDs is illustrated in Figure 2a. Here, we fabricated CsPbBr_3_ QDs films on rigid conductive ITO substrates for conveniently optimizing the detector performance. The CsPbBr_3_ QDs films formed via centrifugal casting served as the photoactive layer and the ITO glass with a slit was used as the two electrodes. In the process of the synthesis of CsPbBr_3_ QDs, oleic acid (OA), Octadecene and oleyamine (OAM) were used as capping ligands. Figure 2b shows the FTIR spectrum of CsPbBr_3_ QDs. There are rotational or vibration absorption peaks at 2920 cm^−1^, 2850 cm^−1^, 1730 cm^−1^ and 1460 cm^−1^, among which 2920 cm^−1^ and 2850 cm^−1^ can be regarded as vibration absorption peaks of C-H single bond. While 1730 cm^−1^ can be regarded as the stretching vibration absorption peak of C=O, 1460 cm^−1^ can be inferred as the deformation vibration absorption peak of NH_2_. The existence of organic ligands can effectively stabilize QDs and improve film-forming property. On the other hand, the excessive capping ligand also makes an insulating layer outside perovskite QDs, which inevitably restricts the charge carrier transfer efficiency for perovskite QDs film. Ligand engineering is needed to balance surface passivation and carrier transport of CsPbBr_3_ QDs [21,26].

Here, we demonstrated a solvent-washed method to control the surface ligand density of CsPbBr_3_ QDs and a balance surface passivation, and carrier transport is obtained via solvent treatment. Washing solvents with different polarities and washing times were optimized. Different solvents with different polarities will result in different elutive power to the ligand on CsPbBr_3_ QDs which affects the morphology and carrier transport of CsPbBr_3_ QDs. Firstly, ethyl acetate, isopropanol and acetone were used as solvents to wash CsPbBr_3_ QDs, and the photoelectric performance of the corresponding detectors was tested as shown in Figure 2c. The switching ratio of the photodetector washed by isopropanol was as high as 384, which was much higher than that of detectors washed by ethyl acetate and acetone (78 and 1.35, respectively). To illustrate the effect of different solvent treatments, the photographs of CsPbBr_3_ QDs dispersed in n-hexane solution under daylight and 365 nm UV light are shown in Appendix A, respectively. The order of Appendix A are the cases without treatment and washed with ethyl acetate, isopropanol, and acetone, respectively. It can be seen that CsPbBr_3_ QDs solutions change from green (clarification) to yellow (turbid). When excited under the UV lamp, we can observe the fluorescent to gradually get dark and the solution washed by acetone is very weak. These phenomena can be explained by the polarity order of the washing solvents (ethyl acetate < isopropanol < acetone). The surface ligand density of CsPbBr_3_ QDs can be controlled by the polarity of solvents. As shown in Figure 2d, acetone with the largest polarity washed away most of the ligands on the surface of QDs. As the organic ligand was washed away, the stability of the QDs gradually weakened, and the aggregation of QDs occurred. Therefore, the solution treated by acetone was turbid and dark; ethyl acetate is the mild one and isopropanol is in the middle. The microscopic morphology of photodetectors treated by different solvents further confirms this conclusion. Appendix A showed the film morphology of the device washed by ethyl acetate and isopropanol is continuous, while the film morphology of the device washed by acetone is poor and crackle. In addition, the washing times of isopropanol were also optimized as shown in Figure 2e,f, and one time provides the best switching ratio. With the increasing number of washes, the solution gradually became turbid from clarification, and the fluorescence intensity gradually disappeared, the film forming morphology of the device also gradually deteriorates. Therefore, detectors washed by isopropanol for one time gets the balance between good morphology and carrier transport of CsPbBr_3_ QDs film. It can also be seen from Appendix A that the optimal switching ratio can be obtained with the device washed once with isopropyl alcohol.

To evaluate the detectivity of CsPbBr_3_ QDs to different incident wavelengths, the I-t and I-V curves of the corresponding photodetectors under different irradiation wavelengths at 40 mW/cm^2^ were recorded in Appendix A and Figure 3a. As expected, the CsPbBr_3_ photodetector is the most sensitive to the light at 405 nm owing to the strong absorption shown in Appendix A, followed by 532 nm, and there is no photoelectric response to light at 660 nm. These results fully demonstrated that the CsPbBr_3_ photodetector is well sensitive to high energy wavelengths. What is more, we also measured I-t and I-V curves at 405 nm as a function of incident light intensity. As shown in Appendix A, the current linearly increased with the incident light intensity. It is because the number of photogenerated carriers is proportional to the absorbed photon flux, which is related to the high light absorptivity of the perovskites. Similarly, it can be found in Figure 3b that the photocurrent gradually increases with the incident light intensity. Appendix A shows that with the increase in voltage, the corresponding photocurrent also increases linearly, which reflects the good contact between CsPbBr_3_ QDs film and ITO glass. In addition, CsPbBr_3_ photodetectors show good stability as shown in Figure 3c. It shows that the switching ratio nearly remained unchanged when CsPbBr_3_ photodetectors were measured for 180 cycles at 40 mW/cm^2^, 10 V. The primary cycle and response time of the photodetector are shown in Appendix A, the rise time is 0.176 s after triggering and the decay time is 0.09 s after termination of irradiation. These response times are as good as other CsPbBr_3_ photodetectors with a similar structure.

### 3.2. Study on the Performance of Paper Based Pressure Sensor

The device structure of the pressure sensor based on PEDOT: PSS-coated cellulose paper is shown in Figure 4a. The porous conductive structure of the PEDOT: PSS-coated cellulose paper is on the top of ITO glass with a slit, and a transparent PDMS film is used to pack the whole device. Generally, when the bias is applied on the two ITO electrodes, the total resistance (R_total_) of the pressure sensor is composed of the bulk resistance of the conductive paper (R_bulk_) and the contact resistance (R_contact_) between the conductive paper and the electrodes. When pressure is applied to the sensor, the bulk is compressed and the contact area increases, and then all these lead to a significant decrease in R_bulk_ and R_contact_. As the deformed part of the pressure sensor, the quality of porous materials greatly affects the device performance. There are several reported ways to conformally grow PEDOT onto porous substrate, such as the oxidative chemical vapor deposition (oCVD), spin-coating and drop coating [27,28]. Here, we select drop coating because it is free oxidant, the preparing process is simple and the content of PEDOT: PSS in the cellulose paper is well controllable. Therefore, we first studied the influence of different papers on the sensing performance of the pressure sensor. The test results are shown in Figure 4b. Compared with spunlace and needled non-woven paper, the sensitivity of the pressure based on PEDOT: PSS-coated cellulose paper is the highest. It is possible that the hydrophobic nature of the spunlace and needled non-woven PET paper may hinder the infiltration of the PEDOT:PSS aqueous solution during the wetting-drying process. In contrast, PEDOT:PSS can readily infiltrate cellulose paper and adhere to the fiber surface. The continuous and uniform conductive network of the cellulose paper exhibits a relative improvement in the sensitivity of pressure sensors.

To further improve the sensitivity of pressure sensors, we optimize the content of PEDOT: PSS in the cellulose paper. Figure 4c shows the influence of different PEDOT: PSS content on the performance of pressure sensors. With the increase in the PEDOT: PSS volume, the optimum volume is obtained at 50 uL. The corresponding device sensitivity reaches 2.64 kPa^−1^ and the response time is 0.406 s (Appendix A). Above 50 uL, the sensitivity of the paper-based pressure sensor began to gradually decrease. From Figure 4d, we can see that the resistance of the conductive paper greatly decreased from 20 uL to 50 uL. After that, the resistance of the conductive paper changes a little. 50 uL is the percolation threshold for the conductive network. Excess PEDOT: PSS may lead to a hard paper and the deformation of the conductive paper will be limited and the detector performance decrease. This result can be further proved by the SEM measurement. Figure 4e–g show the SEM images for cellulose paper, PEDOT: PSS-coated 50 uL cellulose paper, and PEDOT: PSS-coated 150 uL cellulose paper, respectively. It can be seen in Figure 4f that PEDOT:PSS is obviously absorbed on cellulose paper and there is some holes for deformation. However, the cellulose paper is almost fulfilled by 150 uL PEDOT: PSS and the deformation ability of the conductive paper have been greatly limited. Therefore, the optimal PEDOT: PSS content is 50 uL. In addition, the stability of the optimal pressure sensor was measured and shown in Appendix A. During the initial cycles, the value of ΔI/I_0_ gradually decreases, which is due to irreversible deformation of the conductive paper and the loose contact at the interface between conductive paper and ITO glass. After about 300 cycles, the deformation of the conductive paper and the contact at the interface become stable, and then the value of ΔI/I_0_ of the pressure sensor remains stable even after 1000 cycles, indicating that this press sensor based on PEDOT: PSS-coated cellulose paper can provide a continuous and stable sensitivity after the initial transition.

### 3.3. Study on the Performance of Perovskite QDs Dual-Function Sensor

The light/pressure bifunctional sensor is fabricated by a simple superposition combination of the above two kinds of sensors. Its structure diagram is shown in Figure 5a, the photodetector based on CsPbBr_3_ QDs film is at the bottom, and then the PEDOT: PSS-coated cellulose paper was simply stacked up on top of the photodetector. Finally, a transparent PDMS film is used to pack the whole device. In order to evaluate the light/pressure bifunctional sensor, we conduct both optical sensing and pressure sensing tests, respectively. Figure 5b shows the I-t curves when light is from the paper side or the ITO glass side. It shows that the photocurrent from the paper side is lower than that from the ITO side, which is due to the light absorption of PDMS film and conductive paper. The photocurrent from the ITO side is similar with that from the pure photodetector, which indicates that the introduction of conductive paper into the bifunctional sensor nearly has no effect on the performance of the photodetector from the ITO side. At the same time, the pressure sensing performance of the bifunctional sensor was compared with that of the pure pressure sensor. As shown in Figure 5c, the former can also detect the pressure easily. However, the sensitivity has dropped to almost 25% of the pure pressure sensor. We can attribute decrement to the large resistance introduced by the CsPbBr_3_ QDs film, resulting in a smaller current on the pressure sensor and then a smaller ΔI/I_0_ value. Although the pressure sensitivity has declined after compounding, the combined sensor can achieve separate responses to both light and pressure. Therefore, simultaneous responses to both light and pressure are well expected. Figure 5d shows the sensing situation when pressure is applied during a certain illumination. When a UV light was on the bifunctional sensor, the value of ΔI/I_0_ was stabilized at once, and then a periodic pressure was applied. The periodic ΔI/I_0_ responses were observed on the base of the stable photocurrent. It demonstrates that bifunctional sensor can respond to pressure when illuminated and realizes the light/pressure bifunctional sensors were successfully obtained.

### 3.4. Application Research of Flexible Dual-Function Sensor

To explore these bifunctional sensors in the application of e-skin, flexible bifunctional sensors were fabricated. The rigid ITO glass was replaced by the flexible Au interdigital electrodes. The flexible photodetector device based on our CsPbBr_3_ QDs film is illustrated in Figure 6(ai). The continuous and uniform CsPbBr_3_ QDs films can also be formed on flexible Au interdigital electrodes by centrifugal casting. A photograph of the curved CsPbBr_3_ photodetector is shown in Figure 6(aii), demonstrating its high flexibility. We measured the I-V curves using a laser diode at 405 nm, as a function of incident light intensity. It can be seen in Figure 6b that the flexible photoconductive detector based on all-inorganic perovskite CsPbBr_3_ QDs exhibited a high performance. Upon illumination with a 405 nm laser, the typical linear and symmetrical photocurrent versus voltage (I–V) plots indicate that the CsPbBr_3_ QDs were well dispersed on the Au interdigital electrode with good ohmic contact. The demonstrated ohmic contact is due to the energy alignment between the CsPbBr_3_ QDs and the Au interdigital electrode. The current under dark and light conditions for the CsPbBr_3_ QDs photodetector device were 3.32 × 10^−9^ A and 5.6 × 10^−7^ A, respectively, under an applied voltage of 10 V. Therefore, the switching ratio was 168, which indicated that the flexible device showed better light-switching behavior than the rigid device. The main reason is the interdigital electrode structure, which provides more electrode fingers to extract electrons and holes separately.

Similarly, bifunctional sensors based on Au interdigital electrodes are well flexible and shown in Figure 6(aiii,aiv). Benefiting from their bifunctional and flexible properties, our bifunctional sensors as e-skin exhibit promising potential to monitor human motions (Figure 6c). By monitoring the ΔI/I_0_ as shown in Figure 6d, it was observed that the bilayer sensor could accurately detect the flexion of the finger knuckle at different angles (30°, 60° and 90°) and the shape of peaks in curves is almost the same while consecutive bending at the same angles, indicating the excellent repeatability of this e-skin sensor.

To illustrate the light/pressure bifunctional property, we further studied the effects of simultaneous pressure and light stimulation on flexible bifunctional sensors. Figure 6e and 6f show the ΔI/I_0_ vs. time curves of the bifunctional sensor. Figure 6e shows the sensing situation when pressure is applied during a certain illumination. When a 532 nm laser was lit on the bifunctional sensor, the value of ΔI/I_0_ was stabilized at once. Then, different levels of pressure were applied. The value of ΔI/I_0_ increased on the base of the stable photocurrent. On the other hand, Figure 6f shows the sensing situation when illumination is applied during a certain of pressure. When a certain pressure was applied on the bifunctional sensor, the value of ΔI/I_0_ can also stabilized, and then the light was on, and the value of ΔI/I_0_ increased on the base of the stable photocurrent. These results demonstrate that our flexible bifunctional sensor can respond to both light and pressure and further illustrate the great application potential of flexible bifunctional sensors in the flexible multifunctional e-skin for next-generation wearable electronics.

## 4. Conclusions

In conclusion, a UV-blue-green photodetector based on CsPbBr_3_ QDs films was prepared via a simple centrifugal casting method. The n-hexane/isopropanol mixed solvent treatment was conducted to adjust the surface ligand density of CsPbBr_3_ PQDs, which got the balance between good morphology and carrier transport of CsPbBr_3_ QDs film and consequently results in an optimal switching ratio for the photodetector. A novel pressure sensor is prepared by using PEDOT: PSS-coated cellulose paper. It was found that the optimal pressure sensitivity was 2.64 kPa^−1^ when the PEDOT: PSS content was 50 μL. Based on the photodetector and the pressure sensor, we successfully realize light/pressure bifunctional sensors, which can simultaneously respond to light and pressure stimulation. Furthermore, flexible bifunctional sensors were successfully applied as e-skin to monitor light and human motions. They exhibited a high photosensitivity with a switching ratio of 168 under 405 nm light at a power of 40 mW/cm^2^. At the same time, the flexion of the finger knuckle at different angles (30°, 60° and 90°) could be accurately detected. Our work showed that the strategy to introduce perovskite photodetectors into e-skins is feasible and may open a new way for the development of flexible multi-functional e-skin.

## Figures and Tables

**Figure 1 polymers-15-02136-f001:**
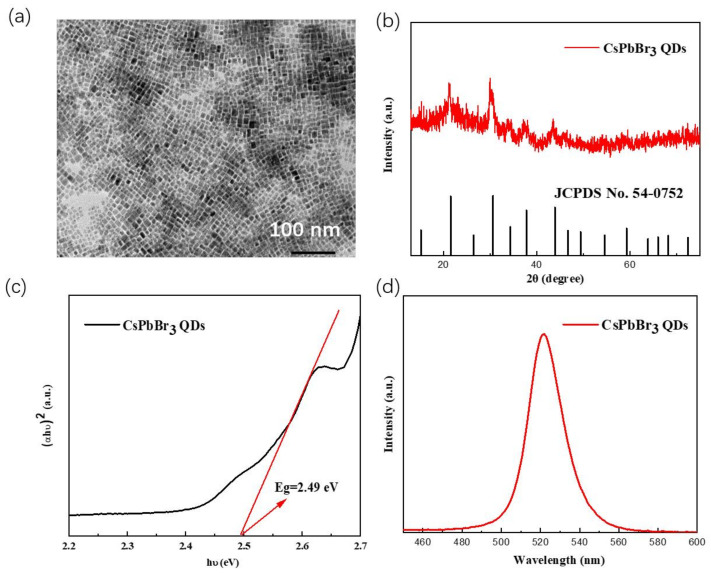
CsPbBr_3_ QDs: (**a**) TEM; (**b**) XRD pattern; (**c**) Optical bandgap, red lines indicate optical bandgaps determined from Tauc plots; (**d**) Fluorescence spectrum.

**Figure 2 polymers-15-02136-f002:**
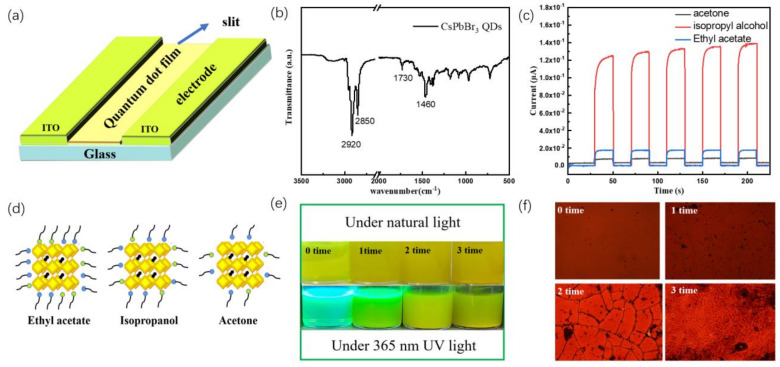
(**a**) Schematic diagram of the PQDs photodetector; (**b**) Infrared spectrum of CsPbBr_3_ QDs; (**c**) I-t curves of photodetectors washed by different solvents under UV lamp (10 v); (**d**) Schematic diagram of CsPbBr_3_ QDs washed by different solvents; (**e**) Micromorphology of devices with different washing times (isopropanol + n-hexane); (**f**) Micromorphology of the device with different washing times.

**Figure 3 polymers-15-02136-f003:**
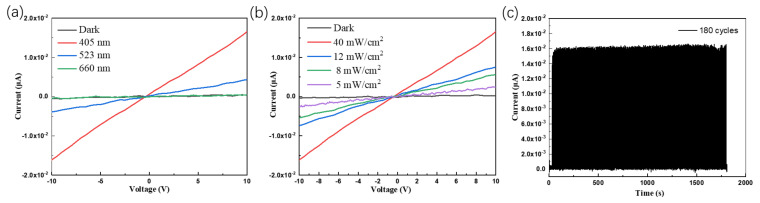
(**a**) Current-voltage curves of photodetectors under different wavelengths (40 mW/cm^2^); (**b**) Current-voltage curves of photodetectors under different optical power density (405 nm); (**c**) Cycle diagram of a photodetector (405 nm, 10 v).

**Figure 4 polymers-15-02136-f004:**
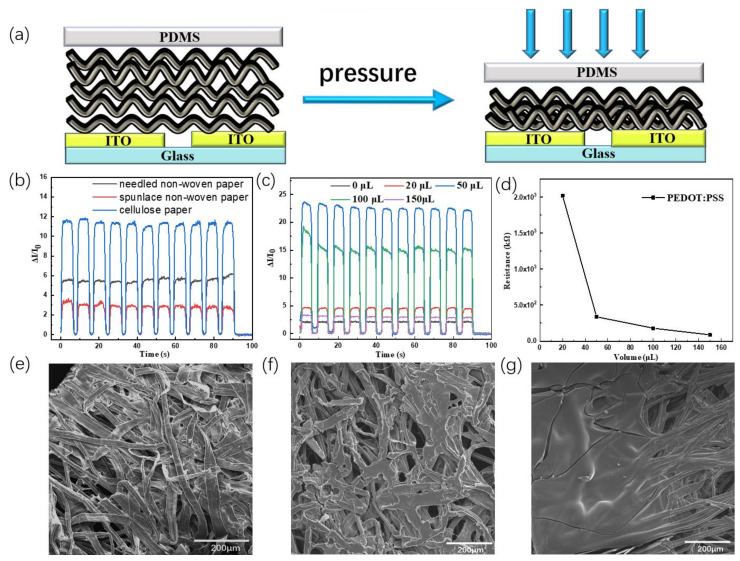
(**a**) Schematic diagram of the pressure sensor and its sensing mechanism; (**b**) The sensing performance of the pressure sensor based on different conductive papers; (**c**) ΔI/I_0_ vs. t curves of the pressure sensor at different PEDOT:PSS concentrations; (**d**) The resistance of the conductive paper with different PEDOT:PSS concentrations; SEM images of (**e**) Cellulose paper; (**f**) PEDOT: PSS-coated 50 uL cellulose paper (**g**) PEDOT: PSS-coated 150 uL cellulose paper.

**Figure 5 polymers-15-02136-f005:**
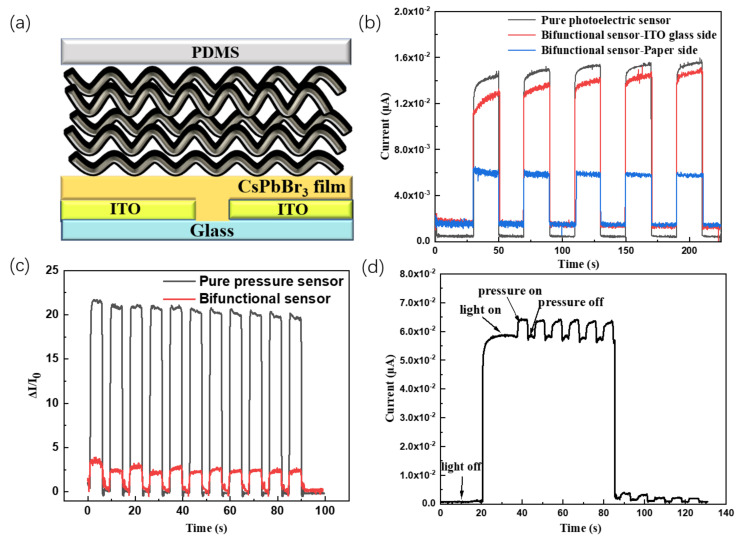
(**a**) Schematic diagram of the bifunctional sensors; (**b**) I-t curves of pure photodetector and the photodetectors lighted at 405 nm from the paper side or the ITO glass side; (**c**) ΔI/I_0_ vs. t curves of the pure pressure sensor and bifunctional sensor; (**d**) I-t curves of light/pressure bifunctional sensor (365 nm); all applied bias is 10 v.

**Figure 6 polymers-15-02136-f006:**
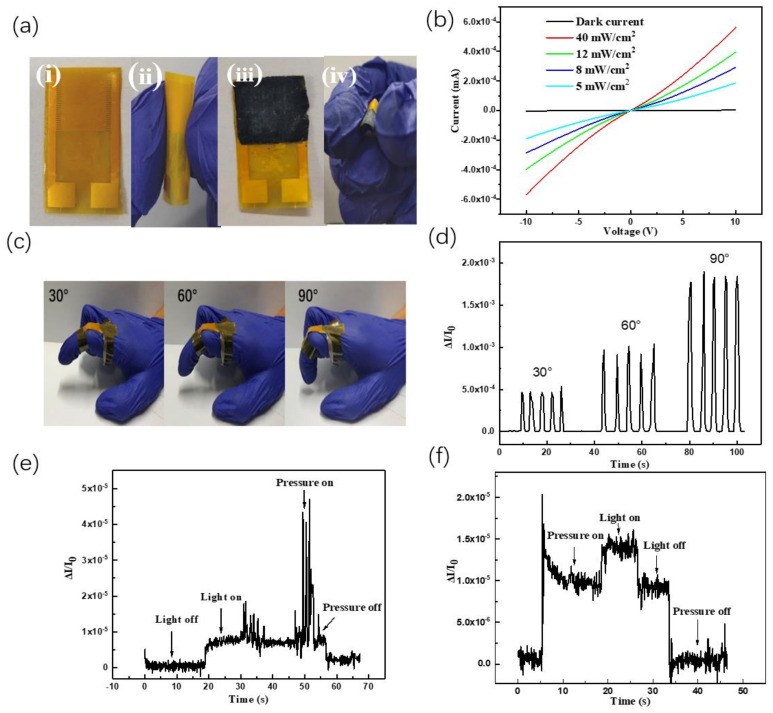
(**a**) Flexible optical sensor (**i**,**ii**) and flexible bifunctional sensor (**iii**,**iv**); (**b**) Current-voltage curves of photodetector under different optical power densities (405 nm); (**c**) Photos and (**d**)ΔI/I_0_ vs. t curves of finger bending at different angles based on flexible bifunctional sensor; ΔI/I_0_ vs. t curves of the flexible bifunctional sensor: (**e**) the force is applied when the light (532 nm) is on and (f) the light (532 nm) is on when is the force applied; all applied bias is 1 v.

## Data Availability

The data presented in this study are available on request from the corresponding author.

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
