# Peer review of "A Light/Pressure Bifunctional Electronic Skin Based on a Bilayer Structure of PEDOT:PSS-Coated Cellulose Paper/CsPbBr_3_ QDs Film"

_polymers, 2023, doi:10.3390/polym15092136_

Round 1

Reviewer 1 Report

The manuscript ID: polymers-2359616, entitled "A Light/Pressure Bifunctional Electronic Skin Based on a Bilayer Structure of PEDOT:PSS-Coated Cellulose Paper/CsPbBr3 QDs Film," was carefully reviewed, and provided comments are listed below. The manuscript reports the investigation on photodetector based on CsPbBr3 quantum dots film and pressure sensor based on PEDOT: PSS-coated cellulose paper. There are good connections between different sections of this review article. However, the manuscript needs further improvement, and I recommend the publication of this manuscript after modifying based on the provided comments.

1)    Page 1, line 16. Please correct the grammar of that sentence. ..based on PEDOT: PSS, not base on……

2)    Figure 1. Please provide the high resolution TEM with the X-ray diffraction. The provided image does not provide any new information and it looks like an SEM.

3)    Page 6, section 3.2. Authors need to provide an SEM image relates to the deposited PEDOT:PSS on paper.

4)    Page 6, section 3.2. Authors must provide information about other deposition method of PEDOT and the advantage of those methods in providing highly uniform and conformal coatings compared to the spin-coating PEDOT:PSS. The below articles based on oCVD method need to be referenced and explained the conformal coatings provided by them:

1)    M. Heydari Gharahcheshmeh, K. K. Gleason, Recent Progress in Conjugated Conducting and Semiconducting Polymers for Energy Devices, Energies, 15, 3661 (2022).

2)    Y. Y. Smolin, M. Soroush, K. K. S. Lau, Influence of oCVD Polyaniline Film Chemistry in Carbon-Based Supercapacitors, Ind. Eng. Chem. Res. 2017, 56, 21, 6221–6228.

Author Response

Dear reviewer:

Thank you very much for your time and useful advices which are very helpful and improve the quality of our manuscript. We hope that the following responses address your questions and suggestions. Please find it in the attachment.

Best regards,

Tengling Ye

Reviewer 2 Report

The manuscript of Wenhao Li et al is devoted to the production of flexible sensory materials based on cellulose. In their work, the authors note the relevance of the study, as a new type of material can change the conductive glass. It is planned to use these materials to collect information about the physiological state of a person. The theoretical part of the manuscript is based on only 18 sources. A total of 25 sources were used in the work. In my opinion, the authors can expand this list and conduct a deeper analysis of already known and published data.
The list of keywords needs to be expanded.
It is imperative to add information about the paper used to the methodological part, as the properties of the resulting material will change depending on the type of paper.
2.3. Characterization - This section should be redone and the methods are detailed.
Figure 2. I suggest that the authors divide this figure into several. This will make the material easier to understand. It is also necessary to improve the quality of the inscriptions in the figures.
Figure 3. The picture quality is poor!
Line 247. Delete extra point.
I recommend that the authors describe the choice of solvent, why was this particular solvent chosen?

Line 247. Delete extra point.

Author Response

(The authors gave the same response as above.)

Reviewer 3 Report

In this article, Wenhao Li et al. investigated the Bifunctional Electronic Skin Based on the Bilayer Structure of PEDOT: PSS-Coated Cellulose Paper/CsPbBr3 QDs Film for dual Light/Pressure detection applications. The interesting development theme will get a wide readership in bifunctional/dual applications. As such, we believe that this paper can be worth publishing in Polymers Journal. However, there are many serious issues that need to be clarified before it can be accepted. For further improvement, there are some suggestions in the referee recommends it be published after the following revisions.

1)      The first two lines in the abstract needs to be modified (line No. 11, 12)

2)      The sentence “helping high-risk workers to see high energy light” should be modified or rewritten.

3)      Give the units of ratio in line No.9.

4)      Some parts of the introduction English are very high to read. Please nominally simplify the sentences without losing the original meaning of the script. The application of the research article is not that much clear. Please check it.

5)      Reference numbers could be differentiated, and authors are suggested to follow the reference formats of Polymer journal.

6)      In the results and discussion part photodetection part with CsPbBr3 QDs, , and pressure sensor with PEDOT: PSS coated cellulose paper showed nice results, and the authors explained the results. But, whereas, the authors have coupled CsPbBr3 QDs/PEDOT: PSS-coated 227 cellulose paper. What is the physical and chemical significance of this structure? In the introduction part, it should be highlighted and also in the abstract.  

7)      The first paragraph of the introduction reveals the importance of e-skin and its consequences. However, it could be better if the authors could add some more information like the methods to fabricate such types of devices and a few literature surveys about these e-skin applications. That further looks better to read the second paragraph. Authors could please consider it.

8)      The sentence “Generally, PFPSs usually” should be modified. In the photodetection case, what is the distance between the source and photodetector? What is the effective area of illumination? Are the light sources used in this research are generate from Laser sources or LEDs, and how did the authors estimate the power densities?

9)      Figure 1d. Depicts only CsPbBr3 QDs, but the authors described “the decrease in thickness of the CsPbBr3 QDs results in a blue-shift of the absorption onset due to quantum size effects”, Please justify it.

10)  Please explain elaborately in the revised manuscript why the photocurrent density is very high in the case of isopropyl cleaning than that of others.

11)  What are the conductivities of the as-prepared materials (PEDOT: PSS, cellulose paper, PEDOT: PSS-coated 227 cellulose paper, CsPbBr3 QDs).

12)  Rather than oxygen in the atmosphere air, have the authors noticed any change in film morphology and photodetection results by the relative humidity effect on the CsPbBr3 QDs films.

13)  How did the authors apply the pressure to the particular sensors, manually or using some automated equipment?

14)  What is the value of pressure? The mechanism of photodetection and pressure sensor is further needing to enrich more for better understanding.

15)  Please provide a comparison table by comparing these nice photodetection results with other published literate, for instance, that includes material, synthesis methods, the wavelength of the light source, etc.

16)  Compare your research work with the latest published photodetectors in various aspects, like structure and parameters. Some of the references used in the manuscript are not up-to-date; authors should cite the following reference doi: 10.1016/j.cej.2023.141473.

Some of the sentences in the manuscript especially ain the introduction is a little hard to understand the concept of this research. Authors should take take care of this. 

Author Response

(The authors gave the same response as above.)

Round 2

Reviewer 1 Report

The manuscript is in a good publication shape and all comments were replied completely.

Reviewer 3 Report

The authors have answered all the queries, and the manuscript is in the right order. Hence, I recommend publishing this article in the Polymers journal with any other comments.